**Data Availability Statement:** Data cannot be shared publicly because of institutional regulations. Data requests are reviewed and approved by the

# The inevitability of Covid-19 related distress among healthcare workers: Findings from a low caseload country under lockdown

Feras I. Hawari[1,2]☯*, Nour A. Obeidat[2]☯, Yasmeen I. Dodin[2], Asma S. Albtoosh[3], Rasha M. Manasrah[2], Ibrahim O. Alaqeel[4], Asem H. Mansour[5]

**1** Section of Pulmonary and Critical Care, Department of Internal Medicine, King Hussein Cancer Center and University of Jordan, Amman, Jordan, **2** Cancer Control Office, King Hussein Cancer Center, Amman, Jordan, **3** Respiratory Division, Internal Medicine Department, University of Jordan, Amman, Jordan, **4** Pulmonologist, Ibn Alhaytham Hospital, Amman, Jordan, **5** Director General Office, King Hussein Cancer Center, Amman, Jordan

☯ These authors contributed equally to this work.
\* fhawari@khcc.jo

## Abstract

### Objectives

To characterize psychological distress and factors associated with distress in healthcare practitioners working during a stringent lockdown in a country (Jordan) that had exhibited one of the lowest incidence rates of Covid-19 globally at the time of the survey.

### Methods

A cross-sectional online survey sent to healthcare practitioners working in various hospitals and community pharmacies. Demographic, professional and psychological characteristics (distress using Kessler-6 questionnaire, anxiety, depression, burnout, sleep issues, exhaustion) were measured as were sources of fear. Descriptive and multivariable statistics were performed using level of distress as the outcome.

### Results

We surveyed 937 practitioners (56.1% females). Approximately 68%, 14%, and 18% were nurses/technicians, physicians, and pharmacists (respectively). 32% suffered from high distress while 20% suffered from severe distress. Exhaustion, anxiety, depression, and sleep disturbances were reported (in past seven days) by approximately 34%, 34%, 19%, and 29% of subjects (respectively). Being older or male, a positive perception of communications with peers, and being satisfied at work, were significantly associated with lower distress. Conversely, suffering burnout; reporting sleep-related functional problems; exhaustion; being a pharmacist (relative to a physician); working in a cancer center; harboring fear about virus spreading; fear that the virus threatened life; fear of alienation from family/friends; and fear of workload increases, were significantly associated with higher distress.

Institutional Review Board at King Hussein Cancer Center (contact Ms. Linda Kateb, at IRBOffice@KHCC.JO). For researchers who meet the criteria for access to confidential data, data can then be shared. Ms. Linda Kateb will be the ultimate provider of the data, following approval by the King Hussein Cancer Center IRB.

**Funding:** The author(s) received no specific funding for this work.

**Competing interests:** The authors have declared that no competing interests exist.

## Conclusion

Despite low caseloads, Jordanian practitioners still experienced high levels of distress. Identified demographic, professional and psychological factors influencing distress should inform interventions to improve medical professionals' resilience and distress likelihood, regardless of the variable Covid-19 situation.

## Introduction

Healthcare practitioners globally are currently facing extraordinary circumstances as a result of the Covid-19 pandemic. From the world's past experiences with other viral outbreaks such as SARS, it is evident that such circumstances impact healthcare practitioners' mental as well as physical well-being, with carry-over effects also being reported even after resolution of outbreaks [1–3]. The experience with Covid-19 is no different, if not more pronounced, due to its being more widespread, and due to the recurring waves of outbreaks, which have had dramatic mental health consequences across various subgroups of the population [4, 5].

In countries across the world, the Covid-19 pandemic has led to heightened anxiety, depression, stress, and insomnia among healthcare practitioners [6–8]. The majority of these countries share the fact that they experienced high caseloads of Covid-19. Conversely, the Kingdom of Jordan in the Middle East represented a differing situation and an interesting case study: the country, at the time of the survey, recorded some of the lowest numbers of cases (in comparison to global numbers) while numbers were surging across the world. The low caseload in Jordan was largely due to stringent measures that were put in place promptly in March of 2020, which included: border closures and limiting free travel; testing and enforced 14-day isolation of all in-bound travelers in designated hotels and hospitals, followed by an additional 14-day quarantine after leaving those hotels or hospitals; imposing a six-week lockdown proceeded by a staggered re-opening of select sectors; banning social gatherings; and restricting the public's movement using a daily curfew [9]. The country only began to experience a surge in Covid cases in mid-September of 2020 when border restrictions were loosened [10].

Like other countries, frontline workers including healthcare practitioners, have been a key component of the country's response plan. Despite their key roles in controlling the outbreak, little has been published about Jordanian frontline workers' experiences and mental health. Specifically in the context of Jordanian healthcare workers, some studies examined knowledge and readiness as it pertains to Covid-19 in pharmacists, dentists and physicians [11–13]. One study examined general anxiety and depression of a national sample inclusive of healthcare practitioners [14]. None have examined in an in-depth manner the prevalence and sources of distress in this group, within its unique local context. Evaluating the predisposition of practitioners to distress, anxiety, sleep and burnout is critical in order to identify mechanisms to address and hopefully alleviate such stress [15, 16]. Importantly, understanding how distress can vary across different scenarios of Covid-19 spread, regardless of caseload, provides valuable information about how healthcare practitioners will potentially respond to the continually changing Covid-19 circumstances across the world.

We sought to evaluate Jordanian healthcare practitioners (physicians, nurses, technicians, pharmacists) fear, distress, anxiety, depression, sleep quality, and fatigue during a period when the country was on high-alert and implementing stringent national measures to control the outbreak. Published studies on healthcare worker distress have been generated from countries with a high caseload. We hypothesized that despite a low caseload, distress, fear and anxiety

would nevertheless be prevalent among healthcare workers as a result of the potential threat of disease emergence or spread. We also hypothesized that key factors, namely, demographics such as age and gender, profession (particularly professions that experienced greater service demand during the outbreak), and workplace environment would be significantly associated with distress. In addition, we measured reported availability of personal protective equipment (PPE) during the country's Covid-19 lockdown. Our study thus aimed to shed light on a low caseload setting and provide a unique perspective on healthcare worker reactions and understand which factors would predispose them to a heightened sense of distress.

## Methods

This study was reviewed and approved by the King Hussein Cancer Center Institutional Review Board (study number 20 KHCC 79), an AAHRPP (Association for the Accreditation of Human Research Protection Programs, Inc) accredited body.

### Study design and sample

A cross-sectional Arabic online survey (https://www.questionpro.com/) was developed and distributed across key governmental and academic hospitals and in community pharmacies largely in the Central region of the country (during lockdown, only hospitals and community pharmacies continued their operations). Distribution channels were purposeful, targeting physicians, nurses, technicians, and pharmacists. Channels included email, text-messaging, and social media groups restricted to healthcare professionals potentially working in these key institutions. The questionnaire was available between April 21, 2020 and May 17, 2020.

### Study variables and measures

The questionnaire (available in a supporting document) was developed and reviewed by a core team of medical staff involved in both research and in Covid-19 screening and potential management. It was composed of the following sections:

1. Mental and general physical health:

   - Distress: our primary outcome of interest was the Kessler distress score [in the past 30 days] [17], which was divided into four categories of no distress (score of 0), low distress (scores of 1 to 5), moderate distress (scores of 6 to 10), and high distress (scores of 11 to 24) [18]. The Kessler 6 scale was selected due to its brevity and reliability, and due to its appropriate reference time period of 30 days, which would have captured most of the lockdown period.

   - Burnout: a validated non-proprietary single-item burnout measure was used to measure burnout. The measure instructs respondents to use their own understanding of burnout and select their level of burnout from five levels (ranging from "I enjoy my work. I have no symptoms of burnout" to "I feel completely burned out and often wonder if I can go on. I am at the point where I may need some changes or may need to seek some sort of help") [19]. Burnout level was dichotomized during the analysis by considering respondents who identified with the third level of burnout "I am definitely burning out and have one or more symptoms of burnout, such as physical and emotional exhaustion" or a greater level, to be suffering from burnout.

   - Anxiety and depression: the Patient-Reported Outcomes Measurement Information System (PROMIS) was used to measure anxiety and depression in the past seven days (PROMIS—Anxiety short-form [20], and PROMIS—Depression short-form [21]). A cut-off of

11 (from a total score of 20) was used to identify at least moderate anxiety or depression. This cut-off was selected because it roughly equated to the T-score that was shown to be a close approximation to other anxiety and depression measure cut-offs [22, 23]).

- Sleep-related issues in the past seven days: three items from the PROMIS sleep-related impairment and the PROMIS sleep impact short forms were used (had a lot of trouble falling asleep; stayed up half of the night at least because you could not fall asleep; and had problems during the day because of poor sleep) [24, 25]; The presence of sleep issues was operationalized as positive if respondents reported trouble falling asleep or staying up half of the night "quite a bit" or "very much" in the past seven days.

- Fatigue in the past seven days: two items from the PROMIS Fatigue short-form were used (felt fatigued; and had trouble starting things because I am tired) [26]. Fatigue was operationalized as positive if respondents reported feeling exhausted "quite a bit" or "very much" in the past seven days.

2. Sources of fear– 21 items covering potential sources of fear due to the Covid-19 outbreak were adapted from other studies that were conducted in comparable situations, namely the SARS outbreak [1, 2]. Two additional items were included to reflect the extent to which respondents were hesitant to go to work or considered resigning. Fear statements were measured using a 5-point Likert scale (from "not at all" to "a very great extent"), and the internal consistency of these items was confirmed (alpha value 0.94). Fear items were then dichotomized for the analysis, by considering those who responded in the highest two points in the Likert scale "to a great extent" and "to a very great extent" as fearful regarding the statement (and all other responses as not exhibiting considerable fear).

3. Workplace characteristics and perceptions about working environment (a selection of items were adapted from published work) [27].

4. Limited access to personal protective equipment (PPE) in the workplace was investigated in our study as a potential source of distress given the global shortage of care resources, including PPE, amid the Covid-19 pandemic [28]. Availability of specific personal protective equipment was measured (items were adapted from a previous SARS-related study) [29]. We explored individual equipment and also created a summary variable, 'PPE availability', which was defined as having access to a mask (surgical or N95), gloves, a gown, and shoe covers.

5. A demographics and professional characteristics section.

## Statistical analysis

Descriptive bivariate statistics were first conducted to characterize levels of distress, fear, anxiety and depression. We specifically focused on examining whether or not distress varied across demographic and professional characteristics, its association with other measures of mental health (such as burnout, fatigue, anxiety and depression), and the potential sources of fear associated with overall distress.

To further understand the ways in which the various mental health related, demographic and professional characteristics were associated with distress, a multivariable analysis was conducted to identify significant factors that were associated with an increased odds of being in a higher distress category. An ordinal logistic regression was used given the nature of our dependent variables (four levels of distress), and model diagnostics were run to ensure that the

multivariable model did not violate the proportional odds assumptions of ordinal logistic regression [30]. The final model included basic demographic and professional characteristics as well as attitudinal measures of fear, work-related experiences, and measures of occupational health (e.g. experiencing sleep issues, exhaustion, or burnout). Although numerous attitudinal and work-related variables were measured in the survey (Tables 1 and 2), we sought to simplify the final multivariate model. Thus, we included only attitudinal and work-related factors that were significantly associated with stress at the bivariate level *and* significantly improved the multivariable model's fit (i.e. variables listed in Tables 1 or 2 and which do not appear in the final multivariable model were not significantly associated with distress after multivariable adjustments, and did not contribute significantly to the final model's fit).

All analyses were conducted in STATA 16 [31, 32].

## Results

Our final sample included 937 Jordanian healthcare practitioners (56.1% females) with a mean age of 33.3 years (ages ranged from 21 to 67). With regards to profession, 68.3% of the respondents were nurses or medical technicians, 13.7% were physicians, and 18.0% were pharmacists. Approximately 42% of respondents worked in a government or academic hospital that provided diagnostic (but not treatment) services for Covid-19; 4.0% worked in a government or academic hospital that provided treatment services for Covid-19; 42.0% worked in a specialized cancer center (which was also authorized to diagnose and refer Covid-19 patients); and 12.0% worked in community pharmacies.

About 20% of the sample suffered from very severe distress (13 or higher Kessler-6 score). When Kessler scores were further categorized into four levels, 32.0% reported high levels of distress (11 or higher Kessler-6 score). Approximately 34% and 19% reported at least moderate anxiety and depression, respectively. In addition, 34.3% of practitioners reported considerable exhaustion; and 28.6% reported having sleep issues (trouble falling asleep or staying up at least half the night). Of those 28.6% reporting sleep-related issues, 55.6% experienced problems functioning during the day because of these.

Detailed descriptive statistics of the sample, in relation to reported levels of distress, are presented in Table 1. Females and respondents falling in the youngest age category were more likely to report higher distress levels (relative to males and respondents falling in the oldest age category); respondents in higher distress level categories were more likely to live with older people, whereas respondents falling in lower distress levels were more likely to be married and have children. Professional and work-related characteristics associated with higher distress included having fewer years of experience, having a Bachelor's degree (relative to having either a lower or higher level degree), working with suspected Covid-19 cases, and experiencing a high workload in the past 30 days. Reporting PPE availability and effective institutional safety measures in the workplace, being satisfied at work, reporting sufficient training in the use of PPE, and reporting positive working relations with peers and co-workers all were significantly associated with being in lower distress categories. Suffering burnout, exhaustion or sleep problems were significantly associated with higher distress levels.

Table 2 displays respondents' perceived fears, cross-tabulated with distress levels. Raw scores for anxiety and depression across distress levels are also displayed. Expectedly, distress levels correlated consistently and significantly with all fear items as well as with anxiety and depression scores. Specific fears that were prevalent included: fear of respondents infecting others (the overwhelming majority, 83.2%, reported this), and fear of families becoming infected in general (65.0% reported this). Conversely, only 31.9% were concerned about themselves being infected. Other sources of fear that resonated with the sample included financial

**Table 1. Demographic, professional and workplace characteristics across distress levels in a sample of Jordanian healthcare practitioners (n = 937).**

| | No distress (n = 29) | Low distress (n = 287) | Moderate distress (n = 321) | High distress (n = 300) | P-value |
|---|---|---|---|---|---|
| **Age (mean)** | 42.1 | 35.8 | 32.8 | 30.7 | < .001 |
| Age category: 30 or younger | 2 (6.9%) | 85 (29.8%) | 136 (42.6%) | 159 (53.0%) | < .001 |
| Age category: 31 to 40 | 14 (48.3%) | 130 (45.6%) | 138 (43.3%) | 120 (40.0%) | |
| Age category: Older than 40 | 13 (44.8%) | 70 (24.6%) | 45 (14.1%) | 21 (7.0%) | |
| **Gender (being male)** | 19 (65.5%) | 158 (55.1%) | 134 (41.7%) | 100 (33.3%) | < .001 |
| **Live with spouse, yes (versus no)** | 24 (85.7%) | 198 (69.7%) | 210 (65.6%) | 160 (53.7%) | < .001 |
| **Have children, yes (versus no)** | 23 (79.3%) | 175 (61.0%) | 184 (57.3%) | 146 (48.7%) | 0.001 |
| **Live with old people, yes (versus no)** | 11 (37.9%) | 119 (41.5%) | 130 (40.5%) | 157 (52.3%) | 0.011 |
| **Live with young people, yes (versus no)** | 26 (89.7%) | 221 (77.0%) | 255 (79.4%) | 246 (82.0%) | 0.254 |
| **Education level** | | | | | 0.002 |
| Diploma or less | 7 (24.1%) | 43 (15.0%) | 35 (10.9%) | 28 (9.3%) | |
| Bachelor degree | 13 (44.8%) | 189 (65.9%) | 244 (76.0%) | 223 (74.3%) | |
| Masters, PhD | 9 (31.0%) | 55 (19.2%) | 42 (13.1%) | 49 (16.3%) | |
| **Occupation** | | | | | |
| Nurses and technicians | 22 (78.6%) | 196 (70.5%) | 209 (66.1%) | 202 (67.6%) | 0.060 |
| Physicians | 6 (21.4%) | 42 (15.1%) | 42 (13.3%) | 36 (12.0%) | |
| Pharmacists | 0 (0.0%) | 40 (14.4%) | 65 (20.6%) | 61 (20.4%) | |
| **Years of experience in the field (mean)** | 17.3 | 11.8 | 9.3 | 7.9 | < .001 |
| **Site of work** | | | | | |
| ICU & ER | 9 (31.0%) | 80 (28.1%) | 90 (28.2%) | 85 (28.6%) | 0.461 |
| Hospital medical departments | 20 (69.0%) | 167 (58.6%) | 180 (56.4%) | 176 (59.3%) | |
| Community pharmacies | 0 (0.0%) | 35 (12.3%) | 46 (14.4%) | 36 (12.2%) | |
| Other (Hospital non-medical departments) | 0 (0.0%) | 3 (1.1%) | 3 (0.94%) | 0 (0.0%) | |
| **Type of institution** | | | | | |
| Specialized hospital (cancer) | 14 (48.3%) | 107 (37.5%) | 139 (43.3%) | 130 (43.8%) | 0.132 |
| Non-cancer/general hospital (government or academic) | 15 (51.7%) | 144 (50.5%) | 135 (42.1%) | 133 (44.8%) | |
| Community pharmacy | 0 (0.0%) | 34 (11.9%) | 47 (14.6%) | 34 (11.5%) | |
| **Exposed to potential COVID patients in line of work, yes (versus no)** | 11 (37.9%) | 128 (44.6%) | 156 (48.6%) | 167 (55.7%) | 0.030 |
| **Work in a Covid-19 specialized ward** | 3 (10.3%) | 50 (17.4%) | 45 (14.0%) | 50 (16.7%) | 0.542 |
| **Experienced a high workload during past 30 days, yes (versus no)** | 5 (17.3%) | 65 (22.7%) | 108 (33.6%) | 137 (45.7%) | < .001 |
| **Was satisfied at work (agree, relative to all other responses)** | 28 (96.6%) | 260 (90.9%) | 235 (73.2%) | 147 (49.2%) | < .001 |
| **Agreed that co-workers could be relied on to do their jobs well** | 20 (69.0%) | 156 (54.6%) | 172 (53.6%) | 138 (46.2%) | 0.037 |
| **Agreed that peers could openly talk about what was and wasn't working** | 25 (86.2%) | 229 (80.1%) | 215 (67.0%) | 147 (49.2%) | < .001 |
| **Agreed that place of work implemented effective safety measures** | 25 (92.6%) | 194 (72.1%) | 189 (62.2%) | 132 (46.2%) | < .001 |
| **Agreed that sufficient training was provided for use of personal protective equipment** | 21 (77.8%) | 160 (59.5%) | 167 (54.9%) | 110 (38.5%) | < .001 |
| **Reported surgical masks were available** | 23 (85.2%) | 220 (81.8%) | 222 (73.0%) | 194 (67.8%) | 0.001 |
| **Reported N95 masks were available** | 17 (63.0%) | 134 (49.8%) | 137 (45.1%) | 91 (31.8%) | < .001 |
| **Reported eye guards were available** | 16 (59.3%) | 120 (44.6%) | 126 (41.5%) | 87 (30.4%) | 0.001 |
| **Reported gowns were available** | 23 (85.2%) | 205 (76.2%) | 191 (62.8%) | 173 (60.5%) | < .001 |
| **Reported gloves masks were available** | 27 (100.0%) | 242 (90.0%) | 269 (88.5%) | 233 (81.5%) | 0.002 |
| **Reported shoe covers were available** | 21(77.8%) | 199 (74.0%) | 193 (63.5%) | 153 (53.5%) | < .001 |

Column total percentages presented (missing values dropped).

**Table 2. Perceived fears and mental health across distress levels in a sample of Jordanian healthcare practitioners.**

| | No distress (n = 29) | Low distress (n = 287) | Moderate distress (n = 321) | High distress (n = 300) | P-value |
|---|---|---|---|---|---|
| **Anxiety, past 7 days raw score (mean)** | 4.1 | 6.3 | 9.1 | 12.5 | < .001 |
| **Depression, past 7 days raw score (mean)** | 4.1 | 4.7 | 6.3 | 11.2 | < .001 |
| **Experienced [quite a bit, very much] sleep disturbances (reference: those who reported some or none)** | 1 (3.5%) | 34 (11.9%) | 71 (22.1%) | 162 (54.0%) | < .001 |
| **Had [quite a bit, very much] fatigue (relative to those who reported some or none)** | 0 (0.0%) | 34 (11.9%) | 91 (28.4%) | 196 (65.3%) | < .001 |
| **Had at least one symptom of burnout (relative to those with no symptoms)** | 1 (3.5%) | 29 (10.1%) | 88 (27.4%) | 196 (65.3%) | < .001 |
| *Fear items* | | | | | |
| High level of fear of being infected | 2 (6.9%) | 49 (17.1) | 102 (31.8%) | 146 (48.7%) | < .001 |
| High level of fear of infecting others | 15 (51.7%) | 211 (73.5%) | 277 (86.3%) | 277 (92.3%) | < .001 |
| Felt virus was close and they were susceptible | 3 (10.3%) | 81 (28.2%) | 135 (42.1%) | 184 (61.3%) | < .001 |
| Felt life was under threat | 0 (0.0%) | 38 (13.2%) | 81 (25.2%) | 152 (50.1%) | < .001 |
| Felt virus was going to go out of control and keep spreading | 1 (3.5%) | 19 (6.6%) | 31 (9.7%) | 99 (33.0%) | < .001 |
| High level of fear of family being infected | 9 (31.0%) | 151 (52.6%) | 208 (64.8%) | 241 (80.3%) | < .001 |
| Felt worried about other health problems | 2 (6.9%) | 24 (8.4%) | 52 (16.2%) | 105 (35.0%) | < .001 |
| Felt worried about family's other health problems | 6 (20.7%) | 111 (38.7%) | 154 (48.0%) | 211 (70.3%) | < .001 |
| Felt worried about their or their family's finances | 8 (27.6%) | 119 (41.5%) | 178 (55.6%) | 216 (72.0%) | < .001 |
| High level of fear of being quarantined | 4 (13.8%) | 63 (22.0%) | 102 (31.8%) | 151 (50.3%) | < .001 |
| Felt worried about family/friends distancing themselves from me due to my job | 0 (0.0%) | 46 (16.0%) | 78 (24.3%) | 148 (49.3%) | < .001 |
| High level of fear of being assigned to a Covid-19 ward | 0 (0.0%) | 42 (14.6%) | 74 (23.1%) | 128 (42.7%) | < .001 |
| Felt reluctant to go to work | 0 (0.0%) | 7 (2.4%) | 31 (9.7%) | 108 (36.0%) | < .001 |
| Felt worried about workload increasing | 1 (3.5%) | 38 (13.2%) | 111 (34.6%) | 191 (63.7%) | < .001 |

Column total percentages presented (missing values dropped).

concerns as a result of the outbreak (55.6%); concerns about other health problems in the family as a result of the outbreak (51.4%); fear about their own susceptibility to the virus (virus is nearing, 43.0%). Approximately 35% were concerned about increasing workloads or being quarantined as a result of the outbreak.

In multivariable ordinal logistic regression results (Table 3), being in the oldest age group and being male continued to be significantly associated with lower distress levels, as were the following factors: having a positive perception of communications with peers (agreed that peers could openly talk about what was and wasn't working), and reporting being satisfied at work. Conversely, suffering from at least one symptom of burnout; reporting functional problems due to sleep-related issues; reporting high level of exhaustion (in the past 7 days); working in a cancer center; harboring fear about the virus spreading uncontrollably; fear that the virus threatened life; fear of alienation from family and friends; and fear of workload increases, were all significantly associated with reporting higher distress levels. The association between being a pharmacist and having a higher level of distress (relative to being a physician) was borderline significant.

## Discussion

Our study evaluates distress levels among healthcare providers in a country that, for several months after the Covid-19 global outbreak, experienced a low caseload due to a stringent lockdown. Our data reveal a high prevalence of fears and distress among healthcare practitioners

**Table 3. Multivariable ordinal logistic regression examining the association between demographic, psychological and professional characteristics on distress level in a sample of Jordanian healthcare practitioners*.**

| | Odds Ratio | p-value | 95% confidence interval | |
|---|---|---|---|---|
| **Age (reference: 30 or younger)** | | | | |
| Age, 31 to 40 | 1.00 | 0.980 | 0.71 | 1.43 |
| Age, older than 40* | 0.59 | 0.030 | 0.37 | 0.96 |
| **Male gender (reference: female)*** | 0.51 | < .001 | 0.37 | 0.69 |
| **Married (reference: unmarried)** | 0.78 | 0.160 | 0.55 | 1.11 |
| **Educational level—Bachelors (reference)** | | | | |
| Educational level—Diploma | 0.82 | 0.37 | 0.53 | 1.27 |
| Educational level—Masters | 1.02 | 0.93 | 0.64 | 1.62 |
| **Profession—Physician (reference)** | | | | |
| Profession—pharmacist* | 2.25 | 0.050 | 0.99 | 5.12 |
| Profession—nurse | 0.83 | 0.46 | 0.50 | 1.37 |
| **Type of institution—non-cancer/general hospital (reference)** | | | | |
| Tertiary cancer center* | 1.69 | < .001 | 1.21 | 2.37 |
| Community pharmacy | 0.61 | 0.23 | 0.27 | 1.36 |
| **Worked with potential or suspected Covid-19 patients (reference: those who did not)** | 1.07 | 0.66 | 0.79 | 1.45 |
| **Reported at least one symptom of burnout (reference: reported no symptoms of burnout)*** | 3.16 | < .001 | 2.19 | 4.56 |
| **Had [quite a bit, very much] fatigue (reference: those who reported some or none)*** | 2.40 | < .001 | 1.68 | 3.42 |
| **Experienced [quite a bit, very much] sleep disturbances (reference: those who reported some or none)*** | 2.44 | < .001 | 1.72 | 3.48 |
| **Agreed that they were satisfied with work (reference: those who disagreed or were neutral to the statement)*** | 0.36 | < .001 | 0.25 | 0.52 |
| **Fear that life was under threat (reference: those who reported no fear or little/some fear only)*** | 1.66 | 0.01 | 1.15 | 2.39 |
| **Fear that virus going out of control and continuing to spread (reference: those who reported no fear or little/some fear only)*** | 2.16 | < .001 | 1.35 | 3.47 |
| **Fear of workload increasing (reference: those who reported no fear or little/some fear only)*** | 1.52 | 0.02 | 1.06 | 2.17 |
| **Fear of family/friends distancing themselves from respondent (reference: those who reported no fear or little/some fear only)*** | 1.58 | 0.01 | 1.11 | 2.26 |
| **Agreed that peers could openly talk about what was and wasn't working (reference: those who disagreed or were neutral to the statement)*** | 0.58 | < .001 | 0.42 | 0.81 |
| **Agreed that sufficient training was provided for use of personal protective equipment** | 0.80 | 0.19 | 0.58 | 1.11 |

*p-value ≤ 0.05.

in Jordan during the lockdown; and confirm that even in circumstances where caseloads may be low, and the healthcare sector has not suffered from a severe stretching of resources, distress and anxiety levels can be considerable. About 32% of our sample reported high distress levels during the study period, with roughly 20% falling in the severe distress category. A third and a quarter of subjects, respectively, also reported at least moderate anxiety and depression, while almost a third reported sleep problems and problems in functionality due to sleep issues. These numbers are comparable to other countries facing high caseloads of Covid-19 [6, 33–36]. and suggest that that facing a new and unknown threat, regardless of the number of cases, is itself a source of stress among healthcare practitioners.

Our findings point to specific demographic factors that are strongly associated with reporting high levels of distress. Older age (which in our study was strongly and directly correlated with years of experience) was inversely associated with distress. Conversely, being female was significantly associated with a greater odds of being in a higher distress category. With the exception of a few studies, most studies have demonstrated a similar effect of being female and being younger on higher levels of mental stress [6, 34, 37–41].

With regards to professional settings, factors that correlated with higher distress levels included working in a cancer hospital. Cancer centers are usually associated with high levels of burnout and distress [42]. Likely further aggravating this situation was the heightened concern

regarding the potentially poor prognosis for cancer patients should they acquire Covid-19, and which has now been documented in other studies [43]. Unlike most other studies [7, 8, 41, 44], working in a Covid-19 designated ward or working directly with Covid-19 patients did not significantly correlate with higher distress levels in our multivariable model. This may have been because the country, at the time of the survey, was experiencing a low caseload. However, there are others who have also reported similar results (the presence of distress despite not having direct contact with Covid-19 patients) [45]. These findings thus suggest that facing a general unknown situation and a stringent lockdown, even among practitioners who were not dealing directly with Covid-19 patients, contributed to feelings of distress in these practitioners. The availability of PPE also did not correlate significantly with distress in our multivariable model. This is not surprising, given that the country had not yet experienced a surge in cases and there were no reported shortages of PPE at the time. Conversely, professional factors that correlated with lower distress levels included general satisfaction at work and positive perceptions of communications between co-workers. This is consistent with what others have found about the effect of organizational support on mental well-being amongst practitioners experiencing the Covid-19 pandemic [7, 39, 46].

We also noted the emergence of pharmacists as a relatively distressed healthcare profession (levels of distress among them exceeded those found among other professions, with borderline significance). We had originally hypothesized that pharmacists would experience greater distress, because during the lockdown period, pharmacies continued their operations, and community pharmacies in particular became the only accessible source of some basic healthcare services for the public. Hospital oncology pharmacists (who comprised the majority of hospital pharmacists in our sample) also were working more intensively, because although fewer pharmacists were being employed per shift, pharmacists were delivering outpatient medications to a greater number of vulnerable cancer patients (using delivery services, which in itself may have posed additional stress given a greater number of patients now not being counseled in the normal manner). In addition, a large study on Jordanian pharmacists, revealed perceived knowledge deficits among pharmacists, which may have been another factor contributing to their distress [11]. Basheti et al specifically reported that approximately half of pharmacists felt they had not received sufficient education about epidemics, and roughly 60% stated that the media (rather than a recognized scientific entity) was their primary source of knowledge about Covid-19.

The important association between occupationally-related physical symptoms and heightened distress also was revealed in our study. Practitioners who suffered from burnout at work, physical exhaustion, and sleep issues were 2.5 to 3 times more likely to have higher levels of distress. The study also highlighted specific fears that were associated with higher distress levels, such as fear that the virus was spreading beyond control, fear about being alienated from friends or family, and the fear of a possible increase in workload. Such fears have been noted by others [47]. Furthermore, although not significant in our final model, it is noteworthy that the most widely resonant fear reported by most respondents was fear of infecting others (roughly 83% were concerned about this, whereas only 33% indicated they were concerned about being infected). This is similar to what others have noted [48, 49]. Fear of infecting others was likely more prominent in our sample due to the cultural setting: in Jordan, similar to other countries in the Middle East, long-term care facilities such as nursing homes, skilled nursing facilities, and assisted living facilities, are scarce. Elderly people are usually cared for by their families who typically live with them or live close to them. Furthermore, it is unusual for young unmarried adults to live alone. Thus, it is common to find Jordanian households with both young and old family members (and relatively large family units), likely explaining why the majority of healthcare professionals were concerned about infecting others.

Our study has some limitations. We were interested in capturing various constructs related to distress as well as occupational health using one measurement tool. However, no published tool contained the breadth of constructs we were interested in. We therefore developed our own questionnaire by reviewing and using modified versions (or parts only) of other published tools. In order to develop a final questionnaire with a reasonable length, we employed brief tools (e.g., short-form PROMIS measures and single-item measures such as the burn-out measure); and in the case of constructs such as sleep and fatigue, we used only select items from the short-form PROMIS measures for these constructs. Arguably, this selection may not capture the underlying constructs with the same precision that the original items would have. Furthermore, we were not able to qualitatively examine in an in-depth manner the exact sources of distress among our high-distress sample, and how these interacted with one another within individuals (others in high caseload settings have used interviews in limited samples to detail specific sources of distress [48]). We also speculate that a source of distress for healthcare workers that was not probed in our study was the general experience of the stringent lockdown. Our survey was not designed to specifically measure this, but others have shown that generally experiencing a lockdown and quarantines negatively impacts mental health [50]. Furthermore, our survey was cross-sectional in nature, and did not capture the effect of fluctuations in the general Covid-19 situation on distress. However, it is relevant to note that although there is a possibility that the symptoms we report may have existed prior to the Covid-19 situation, we additionally inquired about whether or not—among those who reported any symptoms of anxiety or depression, and those reporting sleep issues or exhaustion—such symptoms existed pre-Covid-19, and found that less than 17% of respondents reported that they suffered from these symptoms in the same intensity pre-Covid-19.

Despite its limitations, we have been able to collect valuable data on a large and diverse sample of medical professionals representing various healthcare facilities (governmental and academic hospitals including a tertiary cancer center, and community-based pharmacies), and within a critical time period, during and shortly after a lockdown. Given that practitioners in our sample, without yet having experienced Covid-19 surges, appear predisposed to high levels of distress, our results confirm the need to do more with regards to preparing and protecting healthcare practitioners in anticipation of the possibility of future outbreaks of Covid-19, by enhancing their coping and resilience skills with the aim of maintaining their mental and physical well-being. Our results demonstrate that there are specific groups of healthcare professionals to target as well as specific topics to discuss, in order to preempt workers reaching a state of high distress in the medical workplace, thus preparing them to handle the Covid-19 situation with resilience, regardless of the continually changing environment and the potential for caseload changes in the future. For example, the psychological and functional factors that emerged in our analysis are useful in highlighting thoughts as well as concerns which, if expressed by employees in their clinical practice, can prompt leaders in the workplace to take notice and explore the possibility of distress as well as attempt to alleviate it early on. Work place initiatives such as continually tracking burnout as well as functionality due to sleep issues or exhaustion, and developing programs that foster a positive working environment, may be of value in preempting healthcare workers reaching high levels of distress.

## Author Contributions

**Conceptualization:** Feras I. Hawari, Nour A. Obeidat.

**Data curation:** Nour A. Obeidat, Yasmeen I. Dodin, Asma S. Albtoosh, Rasha M. Manasrah, Ibrahim O. Alaqeel, Asem H. Mansour.

**Formal analysis:** Nour A. Obeidat, Yasmeen I. Dodin.

**Investigation:** Feras I. Hawari, Asma S. Albtoosh, Rasha M. Manasrah, Ibrahim O. Alaqeel, Asem H. Mansour.

**Methodology:** Feras I. Hawari, Nour A. Obeidat, Yasmeen I. Dodin.

**Project administration:** Feras I. Hawari, Yasmeen I. Dodin, Asma S. Albtoosh, Rasha M. Manasrah, Ibrahim O. Alaqeel, Asem H. Mansour.

**Supervision:** Feras I. Hawari, Nour A. Obeidat.

**Validation:** Yasmeen I. Dodin.

**Writing – original draft:** Feras I. Hawari, Nour A. Obeidat, Yasmeen I. Dodin, Asma S. Albtoosh, Rasha M. Manasrah, Ibrahim O. Alaqeel, Asem H. Mansour.

**Writing – review & editing:** Feras I. Hawari, Nour A. Obeidat, Yasmeen I. Dodin, Asma S. Albtoosh, Rasha M. Manasrah, Ibrahim O. Alaqeel, Asem H. Mansour.

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
