## [Decision Letter · Decision Letter 0]

13 Jan 2021

PONE-D-20-18677

The inevitability of Covid-19 related distress among healthcare workers: findings from a low caseload country under lockdown

PLOS ONE

Dear Dr. Hawari,

Thank you for submitting your manuscript to PLOS ONE. After careful consideration, we feel that it has merit but does not fully meet PLOS ONE’s publication criteria as it currently stands. Therefore, we invite you to submit a revised version of the manuscript that addresses the points raised during the review process.

The manuscript has been evaluated by two reviewers, and their comments are available below.

The reviewers have raised a number of concerns that need attention, and they request additiona information on methodological aspects of the study including the reported measures as well as revisions to the statistical analyses.

Could you please revise the manuscript to carefully address the concerns raised?

 A rebuttal letter that responds to each point raised by the academic editor and reviewer(s). You should upload this letter as a separate file labeled 'Response to Reviewers'. A marked-up copy of your manuscript that highlights changes made to the original version. You should upload this as a separate file labeled 'Revised Manuscript with Track Changes'. An unmarked version of your revised paper without tracked changes. You should upload this as a separate file labeled 'Manuscript'.

We look forward to receiving your revised manuscript.

Kind regards,

Vanessa Carels

Staff Editor

PLOS ONE

Journal Requirements:

2. Please improving statistical reporting and refer to p-values as "p<.001" instead of "p=.000". Our statistical reporting guidelines are available at " ext-link-type="uri" xlink:type="simple">https://journals.plos.org/plosone/s/submission-guidelines#loc-statistical-reporting"

Reviewers' comments:

Reviewer's Responses to Questions

**Comments to the Author**

1. Is the manuscript technically sound, and do the data support the conclusions?

Reviewer #1: Yes

Reviewer #2: Partly

2. Has the statistical analysis been performed appropriately and rigorously? 

Reviewer #1: Yes

Reviewer #2: No

3. Have the authors made all data underlying the findings in their manuscript fully available?

Reviewer #1: Yes

Reviewer #2: Yes

4. Is the manuscript presented in an intelligible fashion and written in standard English?

Reviewer #1: No

Reviewer #2: Yes

5. Review Comments to the Author

Reviewer #1: The present study aimed at investigating psychological distress and associated factors in a sample of 1006 Jordanian healthcare workers during a stringent lockdown due to COVID-19 pandemic. Results showed 32% of subjects reporting high distress levels and 20% of subjects suffering very severe distress. Further, around a third and a quarter also reported moderate to severe anxiety and depressive symptoms, respectively, besides another third reporting sleep problems. Authors also reported demographic and other personal characteristics associated to higher distress levels. Particularly, younger age, being female, being pharmacist, working in a cancer hospital, higher workload, suffering burnout or sleep problems, feeling frightened by the virus spreading uncontrollably and by the separation from family and friends were the strongest predictors of higher distress levels.

The topic is timely and well structured. However, some considerations that would help to increase the quality of the work could be taken into account.

Comments:

1) The paper definitely needs the revision of a qualified English native speaker.

2) Authors should provide details on the validity and reliability of each instrument used in the present study. Authors stated that they used only few items of PROMIS questionnaires to assess sleep related impairment and fatigue. The lack of a specific and complete questionnaire to assess these symptoms could represent a limitation when interpreting the present results and should be declared. Further, it could be useful to report which items of these questionnaires were used to the present assessment, describing them in the Methods section.

3) Moreover, burnout was evaluated by a single-item measure, referring to Dolan et al. study (2015). Firstly, the item and its score should be described in Methods. Secondly, the use of a single-item measure could represent another possible limitation in the quality of burnout symptoms assessment, as reported also by Dolan et al. (2015) in their study, and this should be declared in limitations.

4) The paper is interesting because reported data on healthcare workers in a country with low caseloads, confirming the potentially traumatic impact of COVID-19 pandemic on mental health of such population. These data seem to be in line with a recent study assessing burnout, anxiety and depressive symptoms and their association in healthcare workers facing the first phase of COVID-19 pandemic in an Italian region in lockdown with low caseloads (see doi: 10.3390/ijerph17176180). These data could suggest that facing a new and unknown threat was the most stressful factor itself, rather than the number of caseloads. This could be commented when discussing results.

5) Results showed reporting effective institutional safety measures in workplace as well as feeling satisfied at work were significantly associated to lower distress levels. Authors could better discuss this point, for example considering a recent systematic review on healthcare workers facing Coronavirus outbreaks (including COVID-19 pandemic studies) reporting perceived safety of the working environment and a good work organization, as factors which seem to protect healthcare workers to the development of work-related posttraumatic stress, as well as a clear communication of directives and supervisors’ and colleagues’ support (see doi: 10.1016/j.psychres.2020.113312). Similarly to the present results, the same study reported as the fear of infecting others, as well as social isolation and family separation where related to higher distress symptoms.

Reviewer #2: The main aim of this paper was to evaluated distress levels in healthcare workers during the covid-19 pandemic in Jordan. Moreover, the specific aim was to investigate the factors associated with psychological distress in this population in order to carry out psychological interventions to improve medical professionals’ resilience.

Although the topic is interesting, there are some unclear points that must be clarified. The reviewer hopes that the comments below will be helpful to improve the manuscript. The following suggestions are divided into parts.

Introduction

I would suggest you to improve this section by adding other studies about this topic.

Study variables and measures

Referring to the following sentence: “Fear statements were originally measured using a 5-point Likert scale (from “not at all” to “a very great extent”) and then dichotomized for analysis, by considering those who responded “to a considerable extent” and “to a very great extent” as fearful regarding the statement.”, it is not clear the choice to exclude subjects who do not experience fear. Please clarify this point.

Statistical analysis

This section is unsatisfactory. A more detailed description of all analysis carried out is recommended.

Moreover, the authors, have declared the intention to maintain model parsimony. Despite this, I have many concerns about the solidity of the model. Indeed, a high number of variables has been included in the multivariable ordinal logistic regression.

Finally, the latest analysis (the evaluation of the Access to PPE and other perceptions related to PPE use across different professions) deviate from the main objective and risk burdening the study. I suggest to remove them.

Results

The authors point out that the category of nonmedical personnel has been excluded from the analysis. It is therefore not clear why it was included in the final sample. I suggest to take into account the possibility of excluding this category from the final sample.

The authors declared that: “Distress levels correlated consistently and significantly with all fear items and with anxiety and depression scores”.

The association between distress and anxiety and depression scores seems obvious considering that they represent very overlapping constructs. Have the authors taken this into account?

Discussion

I would suggest you to improve this section. The results should be better compared to the literature. There are in fact several other studies about the distress on healthcare workers. Please, see for example Di Tella et al.2020; Castelli et al. 2020.

I suggest you to move the following sentence “Our results suggest that we need to do more with regards to preparing and protecting our healthcare practitioners in anticipation of the realistic possibility of a future surge in Covid-19, given that our finding suggest that our practitioners are already predisposed and have experienced considerable psychological stress.” to the end of the section and to better argue the clinical implications of the study.

Referring to the following sentence “We had originally hypothesized that all pharmacists would experience greater distress…”, it is not clear why the findings are not discussed in relation to the literature (e.g. Basheti et al.)

6. PLOS authors have the option to publish the peer review history of their article (what does this mean?). If published, this will include your full peer review and any attached files.

Reviewer #1: No

Reviewer #2: No

---

## [Author Response · Author response to Decision Letter 0]

8 Feb 2021

Please see attached "Response to Reviewers" document.

---

## [Editor Report · Decision Letter 1]

24 Feb 2021

PONE-D-20-18677R1

The inevitability of Covid-19 related distress among healthcare workers: findings from a low caseload country under lockdown

PLOS ONE

Dear Dr. Hawari

Thank you for submitting your manuscript to PLOS ONE. After careful consideration, we feel that it has merit but does not fully meet PLOS ONE’s publication criteria as it currently stands. Therefore, we invite you to submit a revised version of the manuscript that addresses the points raised during the review process.

We look forward to receiving your revised manuscript.

Kind regards,

Annunziata Romeo

Academic Editor

PLOS ONE

Journal Requirements:

Additional Editor Comments (if provided):

Dear authors,

first of all, I consider it honest to report that I participated as a reviewer for the initial evaluation of this manuscript. I think you have correctly addressed all reviewers' concerns and now the manuscript appears clearer.

Notwithstanding this, I encourage you to pay attention to some grammatical/typographical errors (see for example line 291). Moreover, I suggest you to eliminate the following sentence “which eventually did take place during the months of September through November of 2020 in Jordan”(lines 468-469).Finally, since the literature on this subject is getting rich quickly, I suggest you to include this recent review doi.org/10.3389/fpsyg.2020.569935
---

## [Editor Report · Decision Letter 2]

5 Mar 2021

The inevitability of Covid-19 related distress among healthcare workers: findings from a low caseload country under lockdown

PONE-D-20-18677R2

Dear Dr. Feras Hawari,

We’re pleased to inform you that your manuscript has been judged scientifically suitable for publication and will be formally accepted for publication once it meets all outstanding technical requirements.

Kind regards,

Annunziata Romeo

Guest Editor

PLOS ONE

---

## [Editor Report · Acceptance letter]

23 Mar 2021

PONE-D-20-18677R2 

The inevitability of Covid-19 related distress among healthcare workers: findings from a low caseload country under lockdown 

Dear Dr. Hawari:

I'm pleased to inform you that your manuscript has been deemed suitable for publication in PLOS ONE. Congratulations! Your manuscript is now with our production department. 

Kind regards, 

on behalf of

Dr. Annunziata Romeo 

Guest Editor

PLOS ONE